# COMPARING SUPERVISED LEARNING DYNAMICS: DEEP NEURAL NETWORKS MATCH HUMAN DATA EFFICIENCY BUT SHOW A GENERALISATION LAG

**Lukas S. Huber**
University of Bern & University of Tübingen

**Fred W. Mast**
University of Bern

**Felix A. Wichmann**
University of Tübingen

## ABSTRACT

Recent research has seen many behavioral comparisons between humans and deep neural networks (DNNs) in the domain of image classification. Often, comparison studies focus on the end-result of the learning process by measuring and comparing the similarities in the representations of object categories once they have been formed. However, the process of how these representations emerge—that is, the behavioral changes and intermediate stages observed during the acquisition—is less often directly and empirically compared.

Here we report a detailed investigation of the learning dynamics in human observers and various classic and state-of-the-art DNNs. We develop a constrained supervised learning environment to align learning-relevant conditions such as starting point, input modality, available input data and the feedback provided. Across the whole learning process we evaluate and compare how well learned representations can be generalized to previously unseen test data.

Comparisons across the entire learning process indicate that DNNs demonstrate a level of data efficiency comparable to human learners, challenging some prevailing assumptions in the field. However, our results also reveal representational differences: while DNNs' learning is characterized by a pronounced generalisation lag, humans appear to immediately acquire generalizable representations without a preliminary phase of learning training set-specific information that is only later transferred to novel data.

## 1 INTRODUCTION

Representational alignment is the process of comparing and harmonizing the internal representations of different information processing systems such as deep neural networks (DNNs) and humans (Sucholutsky et al., 2023). Two aspects are crucial for this endeavour: First, representations need to be measured across systems to enable comparison.[1] Second, when disparities are uncovered, the focus shifts to increasing alignment by updating the representation of at least one of the compared systems. To do the latter effectively, the sources of the observed representational differences have to be identified first. Which aspects of the system or the system's input data have to be altered to increase representational alignment? Finding the right aspects is arguably a non-trivial task since the search space of differences between systems like DNNs and humans is vast, and not all of these variations are pertinent to the representations themselves.

Here we argue that to better pinpoint the origin of representational differences—and thus to provide the basis for increasing alignment—it may be crucial to align and compare the process in which representations are acquired. For each learning condition aligned during representation acquisition, the

---

[1]Either this measurement is directly comparable or needs to be bridged into a shared space to allow for comparison.

search space for possible sources of differences becomes more constrained: Eliminating potential relevant causes for representational divergence during the learning process enables attempts to increase alignment to be more data-driven and directed, and thus more effective.

## 1.1 BACKGROUND AND RELATED WORK

One way of identifying representational differences (or similarities) between humans and DNNs is by measuring and comparing their behavior in image classification tasks. [2] For an overview, see Wichmann & Geirhos (2023) and Sucholutsky et al. (2023), section 4.3.4. However, DNN-to-human comparisons are often fraught with difficulty (Funke et al., 2021; Firestone, 2020): Unlike DNNs, which typically learn from scratch using static, uni-modal data, humans process continuous, multi-modal information and leverage prior knowledge. Additionally, while DNNs are predominantly trained in a supervised manner, human learning heavily relies on interactions with unlabeled data[3]. A further central caveat is that comparative studies often focus only on the end-result of the learning process once representations have been formed (e.g., see Geirhos et al., 2018b; Muttenthaler et al., 2022; Dujmović et al., 2020; Zhou & Firestone, 2019; Kubilius et al., 2016). Only a few studies try to directly align the process in which representations are acquired in humans and DNNs.

In one extensive study investigating adult learning, human participants had to learn to differentiate pairs of novel objects (Lee & DiCarlo, 2023). Concurrently, they compared human learning with DNN models consisting of a fixed encoding stage and a tuneable decision stage. The decision stage was adapted based on feedback similar to that received by human subjects. While certain models produced relatively accurate predictions of human learning behavior, they showed weaker one- and few-shot generalisation capabilities compared to human learners. In another study, Orhan (2021) conducted a scaling experiment and found that current self-supervised visual representation learning algorithms require orders of magnitude more training data than humans to achieve similar performance in complex and realistic object recognition tasks. Another way to investigate and compare the process in which representations are acquired is by directly contrasting children's learning with DNNs. Huber et al. (2023) reported that children show robustness to image distortions with significantly less data exposure than DNNs. All these findings emphasize the efficiency of human learning processes over current DNNs. Here we aim to contribute to this emerging body of research by providing a side-by-side comparison of learning dynamics in supervised visual representation learning and assessing if these discrepancies persist when relevant aspects of the learning process are carefully controlled.

## 1.2 PRESENT STUDY

In the present study, we introduce a psychophysical paradigm that allows for the direct comparison of not only the end-results of visual representation learning but also its dynamics, from the initial absence of representation to the availability of generalizable representations. More specifically, we measure and compare how humans and various DNNs acquire representations for novel objects in a constrained environment with aligned learning conditions. We track behavioural changes across six epochs of a non-trivial image classification task—reflecting the train-test iteration process common in machine learning. At each epoch we measure and compare data efficiency and how well learned representations can be generalized to previously unseen data. During representation acquisition we align the following aspects of the learning process:

**Learning goal.** Both humans and DNNs are tasked with learning representations of 3D objects that were previously unknown to them.

**Starting point & prior knowledge.** In contrast to DNNs, which often begin learning from scratch, adult humans are equipped with a wealth of representations of natural and artificial objects. To level the playing field at least somewhat, we utilize DNNs pre-trained on ImageNet, hopefully mimicking at least partially the extensive representational knowledge adult humans possess already. Given that human and DNN performance on original ImageNet images is usually near-ceiling (Geirhos et al., 2018b; Russakovsky et al., 2015; Shankar et al., 2020; Dodge & Karam, 2017), it suggests that these

---

[2]This study focuses on supervised learning of visual representations. However, note that there is substantial research on human-to-machine comparison in few-shot learning (e.g., Lake et al., 2013; Morgenstern et al., 2019) and reinforcement learning (e.g., see Eckstein & Collins, 2020; Mnih et al., 2015; Silver et al., 2016).

[3]For a comprehensive discussion of the issues arising when using DNNs as models of human vision see Bowers et al. (2023) and the accompanying peer commentaries.

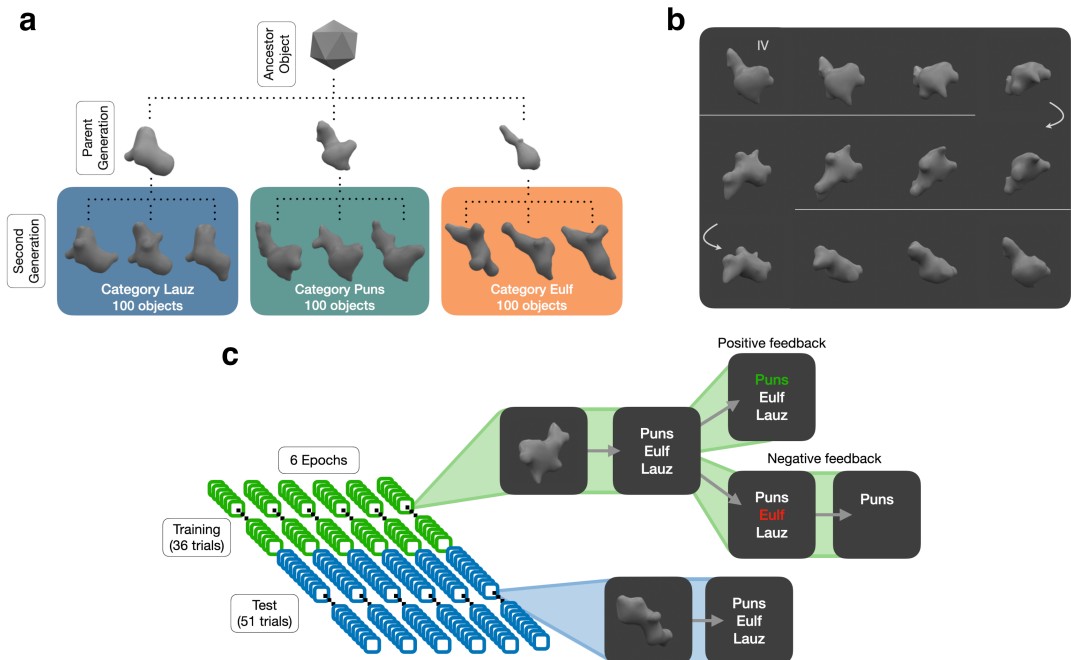

Figure 1: **Naturalistic novel objects were generated to serve as the basis for the creation of stimuli employed in the learning task**. (**a**) Taxonomy diagram illustrating the genesis of novel objects, originating from an icosahedron and spanning two generations, with the second generation serving as the basis for our stimuli. Objects sharing the same parent in the generation process are assigned to the same category. (**b**) Displayed is a set of twelve distinct renderings of an object from the category "Puns", showcasing 30-degree pitch rotations (x-axis) from its initial position (indicated by "IV"). Panel (**c**) shows a blueprint of the learning task. There are six epochs each consisting of 36 training trials (green) and 51 test trials (blue). All test sets contain different images to ensure novelty. During training, participants received corrective feedback mimicking supervised learning in DNN training. Depending on whether the given response was correct or not, the chosen category would either light up in green or red for one second. If the response was incorrect, this would be followed by a screen displaying the correct response for one second. In test trials, no feedback is provided, aiming to restrict supervised learning to the training phase.

pre-trained models and humans are starting from a similar level of existing image classification capacity for i.i.d data.

**Input data.** In typical learning scenarios, humans learn from continuous, multi-modal data (e.g., see Zaadnoordijk et al., 2022), whereas many DNNs are trained on static and uni-modal input. Here we eliminate this difference in learning conditions and match the input data such that both systems are only provided with the exact same static 2D images.

**Learning modality.** It has been argued, that humans do not typically learn from labelled data, but rather in a semi- or unsupervised manner (e.g., see Gibson et al., 2013). Consequently, it has been put forward that self-supervised models are a better proxy for human learning (e.g., see Lotter et al., 2020; Konkle & Alvarez, 2022). However, for behavioral outcomes in image classification tasks Geirhos et al. (2020) has not found a better alignment of self-supervised models with human observers as compared with supervised models (but see Storrs et al., 2021; Zhuang et al., 2021). Reflecting on these findings, our paradigm adopts a supervised learning environment for both human participants and DNNs. This approach facilitates the alignment of both systems within a consistent and controlled framework.

The resulting side-by-side comparison of learning dynamics enables us not only to ensure that observed differences or similarities in the way that information is represented are not mere artifacts of unaligned learning conditions, but also to systematically investigate presumed differences between humans and DNNs, previously hypothesized to account for the disparities in measured representations.

## 2 METHODS

### 2.1 STIMULI

To track the acquisition of object representations, a suitable dataset of stimuli is needed: Novel objects for which neither humans nor DNNs have representations available yet. However, for maintaining external validity and ensuring the generalizability of results, it is crucial that these objects and categories, despite being novel, still bear relevance to real-world conditions. In other words, objects should be naturalistic and the formation of categories within the dataset should not strictly adhere to arbitrary rule-based definitions but rather reflect the concept of family resemblance (Wittgenstein, 1968, 65ff).[4]

Hauffen et al. (2012) provided a method to generate 3D objects that satisfy these conditions (see also Brady & Kersten, 2003; Pusch et al., 2022). They describe a set of algorithms simulating the biological process of embryogenesis and the evolutionary process of phylogenesis to create novel objects. Here we employed these algorithms to create a dataset suitable to study the acquisition of novel representations in humans and DNNs.[5] Beginning with an icosahedron as an ancestor object, we initially generated a primary set of three distinct objects, forming the parent generation. Subsequently, from each of these parent objects, we created a second generation comprising 100 unique objects per parent: the Lauz, Puns and Eulf. Categories were established based on lineage, defining each category by the objects that shared a common parent in the initial generation (see Figure 1, Panel **a**).

For each of the 3D objects we created a series of 23 images (224×224 pixels) rendered using `Blender`'s Python API (version 3.5.0). This rendering involved rotating each object along the pitch (x) and yaw (z) axis at intervals of 30 degrees, capturing various perspectives. An example of some rotations can be seen in Figure 1, Panel **b**. The process resulted in a stimulus set comprising 6,900 images in total (3 categories × 100 objects × 23 rotations). In a pretest we found that object categories were too similar and participants struggled with learning. Thus, to enhance intraclass coherence and interclass separability, we restricted the dataset to a subset of 3,450 images, focusing on the 50% most similar objects within each category. This selection was obtained using the structural similarity index measure (Wang et al., 2004, known for its typically reasonable alignment with human visual perception of similarity), to calculate pairwise similarities of the objects' initial position renderings within each category.

### 2.2 LEARNING ENVIRONMENT

For behavioral comparisons to be meaningful, we aimed to match and maintain consistency in the learning environment across both humans and DNNs. To this end we adapted the standard train-test protocol—commonly employed to train DNNs—also for human observers. The basis for training and test phases is a *forced-choice, time-limited image classification task*. Each trial of this task is built in the following way: After a fixation stimuli (combination of bulls eye and cross hair, see Thaler et al., 2013) (400 ms), participants are presented with a target-image in the centre of a computer screen for 300 ms, which is followed by a pink noise mask with $1/f$ spectral shape (300 ms). On a subsequent response-screen participants have to choose one of three categories, which they think corresponds to the image just seen (this is a standard procedure to collect psychophysical data, e.g., Huber et al., 2023; Geirhos et al., 2018a).[6] For the training phases, participants receive feedback on whether the given response was correct or not—mimicking supervised learning from labeled data. After being presented with all images of the training set, performance is evaluated on a test set to assess the progress in terms of transfer learning (for an overview see Pan & Yang, 2009). To align with the condition of DNNs having their parameters frozen during evaluation, effectively restricting learning to the training phase, we implemented two adjustments. Firstly, to prevent supervised learning during evaluation, human observers did not receive any feedback for test trials. Secondly, to guarantee that both humans and DNNs always encountered previously unseen images, we ensured each test set was composed of hitherto unseen images. By contrasting the development of training and test errors across several intervals

---

[4]According to Wittgenstein, category boundaries are often not sharp but fuzzy, characterized by a series of overlapping similarities rather than a set of necessary and sufficient features shared by all members—a concept he encapsulates as family resemblance.

[5]See Appendix A for further details.

[6]The short presentation time and the noise mask are employed to ensure the involvement of mainly feedforward mechanisms and thus to measure core object recognition as described in DiCarlo et al. (2012).

of training and testing, it becomes possible to track and compare the acquisition and generalizability of object representations. For an overview of the learning task, see Figure 1, Panel **c**.

## 2.3 DATASET COMPOSITION

We randomly selected six objects per category for the training set, choosing two views for each object (the initial position and a 90-degree rotation on the yaw-axis), resulting in a total of 36 images. To ensure the test sets contained previously unseen images, we included novel perspectives of training objects as well as images of objects not encountered during training. Thus, we included novel perspectives of training objects as well as images of objects not encountered during training. Specifically, each of the six *different* test sets comprised eight novel perspectives (both minus and plus 30 and 60 degrees along the pitch and yaw axes) of one training object, and identical rotations (including the initial viewpoint) of a previously unencountered object from each class, resulting in 51 images per test set, resulting in 51 images per test set ($3 \times 8 + 3 \times 9$). This approach ensured that each test set included unique, previously unseen images and maintained a consistent level of difficulty across test sets by using identical rotations. Additionally, this allowed us to evaluate the ease of transferring learned representations to novel perspectives of familiar objects compared to recognizing familiar perspectives of new objects.

## 2.4 OBSERVERS

**Humans.** We collected data from 12 human participants recruited from the subject-pool of the University of Bern, in a controlled lab setting. In addition to the pool-intern course credit, they also had the chance to win a cafeteria voucher worth \$ 15 if they exceeded a classification performance of 70%. If they did not, they received a can of mate tea instead. Images were presented at the center of the screen corresponding to $5° \times 5°$ of visual angle at a viewing distance of approximately 69 cm. A constant viewing distance was maintained by employing a chin-rest and adjusting the monitor's position to ensure participants did not have to look either up or down. Before iterating through all training and test sets, participants completed ten practice trials where they were familiarized with the task using images of sea lions, whales, and dolphins.

**Models.** We fine-tuned and evaluated two groups of DNN models. The first group consisted of classic convolutional neural networks (CNNs) used for image classification and included: AlexNet (Krizhevsky et al., 2012), VGG-16 (Simonyan & Zisserman, 2014), and ResNet-50 (He et al., 2016). To account for recent developments in computer vision research, we also included several state-of-the-art (SOTA) models with refined architectures and training protocols: a vision transformer (ViT Dosovitskiy et al., 2021), an attention-based network, relaxing the translation-invariance constraint of CNNs; a ConvNeXt (Liu et al., 2022), which is model that incorporates design principles and architectural elements from both CNNs and ViTs; and a second generation EfficientNet (Tan & Le, 2021), a model designed to be more efficient and effective, particularly in terms of speed and accuracy for both training and inference tasks. All models were pre-trained on ImageNet-1K and obtained either from the `PyTorch` model zoo (AlexNet, VGG-16, ResNet-50, ConvNeXt and EfficientNet) or from the `Hugging Face` model library (ViT). Prior to fine-tuning, the output layer of each model was modified to enable predictions across three distinct classes. Input data was not preprocessed and no data augmentation was applied—to keep learning as similar as possible between DNNs and humans. For all models we used a Adam optimizer with a constant learning rate of 0.001, and a batch size of 4.[7] For each model, we conducted twenty fine-tuning runs, spanning six epochs, using the identical training data set as used for the human observers. During training, predictions were continuously logged for each image, and after each epoch, predictions for the respective test set were also logged. Additional details about the employed models can be found in Appendix D.

## 3 ANALYSIS & RESULTS

We aimed to measure and compare the behavioral changes and intermediate stages observed during the acquisition of novel representations in humans and DNNs. To this end, we not only evaluate how well the learned representations can be generalized to previously unseen data (Figure 3) but also calculate

---

[7]Since the test sets contained 51 images, the last batch of test images consisted only of 3 images.

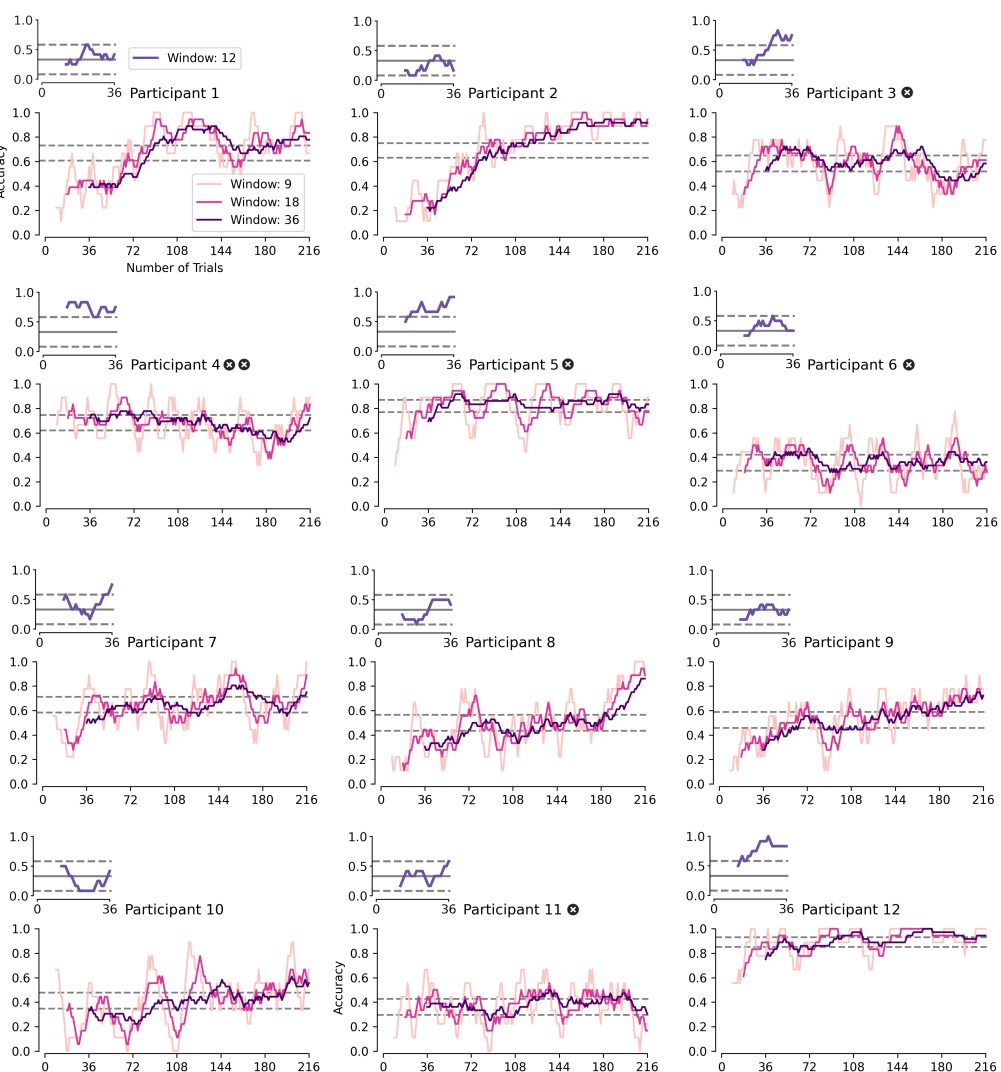

Figure 2: **Inter-individual differences in human learning dynamics.** For each participant we show moving averages of classification accuracy across the whole training (different trajectories mark different window sizes). Early learning is separately displayed in the small plots (purple trajectories). In these plots dashed lines mark binomial (Clopper-Pearson) confidence intervals around chance performance. We reason that if the trajectory does not exceed the upper bound of this confidence interval, this indicates that the observer does start training without any prior knowledge. The larger plots illustrate the evolution of training performance throughout the entire training period, as depicted by moving averages of varying sizes (different shades of pink). Here, dashed lines mark a confidence interval around the mean performance of individual observers. If the performance was significantly below the mean at the start, and is significantly higher at the end of training, this indicates that learning has occurred. Black circles with white crosses mark excluded participants because they either did not start learning from scratch (Participant 4), or did not show any indication of learning (Participants 3–6, and 11).

and compare data efficiency (Figure 4) and generalisation lag (see Table 1 & Figure 5) in humans and DNNs.

## 3.1 ENSURING EQUAL STARTING CONDITIONS AND OCCURRENCE OF LEARNING

To ensure a level playing field, we first confirmed that humans did not possess pre-existing representations of the novel objects. This was assessed by observing the classification performance in the initial training epoch, under the assumption that starting from zero knowledge should result in performance

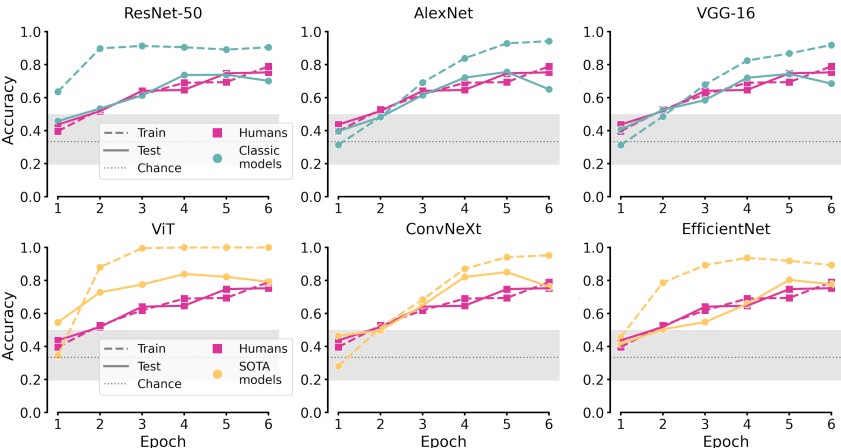

Figure 3: **Observed learning dynamics indicate that both, humans and DNNs, learn novel general-izable representations from limited amounts of training data.** However, while humans immediately form generalizable representations, DNNs show a pronounced generalisation lag. Different plots show training (dashed lines) and test (solid lines) performance trajectories in terms of classification accuracy for all classic CNNs (teal) and SOTA models (yellow). The performance of models is averaged across 20 fine-tuning runs (see Appendix D for individual runs). In each plot, model performance is contrasted to the mean performance of human observers (pink). Detailed information on the learning dynamics of individual participants can be found in Figure 2 (training) and in Appendix B (test). Shaded areas designate binomial confidence intervals (Clopper-Pearson) around chance performance for training sets—accuracies exceeding the upper bound suggest performance significantly above chance.

levels *not* exceeding chance at the onset of training. We evaluated each participant's classification ac-curacy using a moving average over 12 trials, which corresponds to one-third of the training set (see tiny plots in Figure 2, purple colored trajectories). When we assessed whether the initial accuracy surpassed the upper bound of a confidence interval around chance level performance, only Participant 4 started with a classification accuracy significantly above chance level and was thus excluded from further analysis. All other participants initially demonstrated classification accuracies not significantly higher than chance, suggesting no pre-existing familiarity with the categories to be learned. However, during the first epoch, some showed signs of early learning, as indicated by their progression from chance-level accuracy to significantly higher scores by the end of the first epoch.

To evaluate whether the human observers actually learned during the training, we examined the change in training performance across all epochs (see larger plots in Figure 2). For each participants we calculated their mean classification performance across all training epochs. We considered a participant to have learned if they began the training with an accuracy significantly lower than their own mean performance, and then achieved a performance significantly higher than this average by the end of the training. This approach revealed that, of the remaining 11 participants, 7 exhibited clear learning based on this (strict) criterion. Consequently, our subsequent analyses are exclusively focused on this subset of 7 learners who demonstrated tangible improvement over the course of the training without prior knowledge at the beginning of the experiment.

## 3.2 OBSERVING AND COMPARING LEARNING DYNAMICS IN HUMANS AND DNNS

Across humans and all models we consistently observe not only progressive learning but also effective transfer to previously unseen instances in the test datasets. In the subsequent paragraphs, we look closer at the learning dynamics of both humans and DNNs. In Figure 3, we show training and test accuracy as a function of epochs for all models and human observers. Further details can be found in Appendix B, C & D.

**Trainig performance.** All observers, spanning human participants and various model architectures, effectively learned to categorize the training data (see dashed lines in Figure 3). By the end of the sixth epoch, humans demonstrated substantial learning, correctly identifying 78.96% of all training images.

Classic CNNs achieved a mean accuracy of 92.27% by the best training epoch, a performance that SOTA models slightly improved upon, achieving an average accuracy of 96.34%. This close-to-ceiling performance on training data contrasts with the 78.96% accuracy attained by humans. However, this seems not surprising since DNNs tend to memorize training data, particularly when there are only few training samples available (Arpit et al., 2017). Such overfitting to the training data would result in poor generalization to the test sets, but, as detailed in the next paragraphs, that was not the case in the present study.

**Early learning.** To analyse early learning of generalizable representations in the initial epochs, we computed a binomial (Clopper-Pearson) confidence interval centered around chance test performance (assuming $p = 33.\overline{33}\%$ and $n = 51$; size of the test sets). Both humans and DNNs consistently outperformed the upper bound of the confidence interval of 47.05% by the second epoch, underscoring the rapid acquisition of generalizable representations. This early proficiency in learning shared across humans and diverse model architectures highlights the inherent potential to learn generalizable representations in both biological and artificial systems.

**Generalisation.** Increasing test performance across learning indicates that both humans and DNNs successfully transferred the acquired knowledge to previously unseen images, suggesting the emergence of generalizable representations and not mere memorization of the training data (see solid lines in Figure 3). Throughout the training, human participants exhibited a consistent increase in test performance, reaching 75.35% at epoch 6. While all models exhibited steadily increasing test performance until epoch 4, some plateaued at this point and showed a decrease in test performance towards the end of training, indicating signs of overfitting. Consequently, for a fair comparison, only the peak test performance was considered as a measure of transfer learning in models, akin to early stopping. By this metric, classic CNNs reached a test performance of 74.64%, with SOTA models outperforming at 83.13%, placing humans in a similar range to that of classic CNNs.[8]

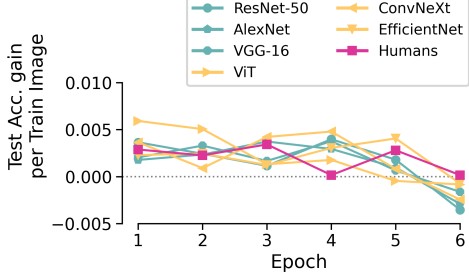

Figure 4: **DNNs are not inherently less data efficient compared to humans.** Here we quantify data efficiency as the mean test accuracy gain per training image across epochs. All observers (humans, classic CNNs, SOTA models) show similar data efficiency across learning, challenging the prevailing assumption that DNNs are inherently less data efficient than humans.

Table 1: **Immediate generalisation in humans and generalisation lag in DNNs.** For models, color reflects group (teal: classic CNNs, yellow: SOTA models). The *Epoch* column specifies which epochs are considered when computing $\Delta G$—the first integer indicates at which epoch training performance significantly exceeds chance level and the second until which epoch test performance increases. If the second integer does not equal 6, this signals overfitting.

| Observer | $\Delta$ G | Epochs |
|---|---|---|
| Humans | 0.002 | 2–6 |
| ConvNeXt | 0.048 | 2–5 |
| VGG-16 | 0.107 | 3–5 |
| AlexNet | 0.122 | 3–5 |
| ViT | 0.178 | 2–4 |
| ResNet-50 | 0.232 | 1–4 |
| EfficientNet | 0.256 | 2–6 |

**Data efficiency.** As quantification of data efficiency, we calculate the average test accuracy gain per additional training image at each epoch (Figure 4). In other words, this metric assesses how effectively a model leverages the training data to enhance its generalisation performance. For each epoch ($i$) the accuracy gain pe training image is given by $\frac{acc_{test,i} - acc_{test,i-1}}{n\ training\ images}$.[9] For both humans and DNNs, data efficiency remains relatively constant

---

[8]We also investigated how well humans and DNNs generalized to previously unseen perspectives of training objects vs. to familiar perspectives of previously unseen objects. The results suggest that there is no significant distinction in performance between these two kinds of test images (see Appendix C).

[9]To calculate the data efficiency at the first epoch, $acc_{test,i-1}$ was assumed to be 33.33% (chance performance).

over the course of learning.[10] Across all epochs, humans, CNNs and SOTA models show comparable data efficiency. These results indicate that—at least in our aligned learning environment—DNNs show a level of data efficiency that is on par with that of humans.

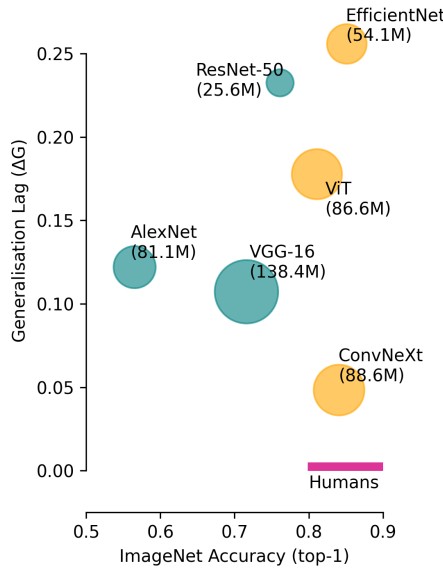

Figure 5: **Better performance does not imply lower generalisation lag.** Generalisation lag ($\Delta G$) is plotted as a function of ImageNet accuracy and number of parameters optimised during training (circle area). Top-1 ImageNet accuracy for humans is somewhat difficult to estimate from available studies since different measures are reported: 92.9–99% top-1 accuracy for entry-level categories (Dodge & Karam, 2017; Geirhos et al., 2018a), 84.9% top-5 accuracy on ImageNet-1k (Russakovsky et al., 2015), and up to 97.3% multilabel-accuracy (Shankar et al., 2020). Therefore, we designate human top-1 ImageNet accuracy with a rather conservative lower and upper bound of 80–90%.

**Generalisation lag.** Despite these similarities in data efficiency, distinct generalisation patterns are observed across the course of learning. Humans succeed in immediately generalizing from the training data to previously unseen images, as indicated by the alignment of their training and test performance trajectories (see Figure 3). Models, however, reveal a marked discrepancy between training and test performance, with test performance consistently lagging behind. To quantitatively assess this generalisation lag ($\Delta G$), we computed the mean difference between training and test accuracy, selectively focusing on epochs where learning occurred—excluding epochs where training performance did not significantly exceed chance level and epochs where models showed signs of overfitting. More specifically, generalisation lag is given by

$$\Delta G = \frac{1}{|E|} \sum_{i \in E} (acc_{train,i} - acc_{test,i})$$

whereby $|E|$ denotes the number of epochs included, $acc_{train,i}$ represents the training accuracy at the $i^{th}$ epoch, and $acc_{test,i}$ is the test accuracy at the $i^{th}$ epoch. The set $E$ includes epochs where $acc_{train,i}$ is significantly above chance and $acc_{test,i}$ has not started to decline, ensuring that only relevant epochs contribute to the $\Delta G$ calculation, thereby providing a standardized measure of generalisation lag. Table 1 summarizes generalisation lag for humans and different DNNs. Comparing the acquisition of representations in terms of $\Delta G$ suggests divergent learning dynamics between humans and DNNs. Humans demonstrate a near-instantaneous ability to generalize, evidenced by a minimal generalisation lag ($\Delta G = 0.002$). On the contrary, models display a substantial generalisation lag, with values ranging from $\Delta G = 0.048$ for ConvNext to $\Delta G = 0.251$ for EfficientNet, illustrating distinct learning strategies between humans and DNNs. Subsequent examination was directed at the potential relationship between generalisation lag and variables such as ImageNet accuracy or model size (Figure 5. Intuitively, one might anticipate an inverse relationship where higher ImageNet accuracy corresponds to a lower generalisation lag. However, our findings revealed no such pattern, indicating that neither ImageNet accuracy nor model size serves as a reliable predictor of generalisation lag.

## 4 DISCUSSION

In the present work we measured and compared the acquisition of novel representations in human observers and DNNs. To ensure the validity of our behavioral comparisons, we carefully controlled the learning environment and parallelized the learning conditions for both participants and models: Both groups were exposed to similar initial conditions and learned exclusively from static, unimodal input data, and similar supervised learning signals. This tailored learning task required the recognition

---

[10]As a result of overfitting at the end of training, the models' data efficiency falls below zero by epoch six.

and categorization of images showing novel naturalistic 3D objects. Objects belonged to one of three classes, each demarcated by fuzzy, rather than distinct boundaries—mimicking real-world categories. Periodic evaluation phases during training examined the ability to transfer learned representations to previously unseen inputs. Our findings reveal that both humans and DNNs have the capacity to learn generalizable representations from minimal training data.

These findings challenge certain prevailing assumptions in the field. It has been frequently posited that human learning is exceptionally data-efficient, showcasing robust one- and few-shot learning capabilities (Lake et al., 2013; Lee & DiCarlo, 2023; Morgenstern et al., 2019), while DNNs, and CNNs in particular, are believed to be less efficient, requiring significantly more training data to reach human-level performance in object recognition tasks (Huber et al., 2023; Orhan, 2021). Contrary to these assertions, our results indicate that when learning conditions are equalized and within a supervised learning context, this disparity diminishes.

While our findings suggest that humans may not inherently possess an advantage in data efficiency, we identified notable representational differences, particularly in terms of generalisation lag. Our analysis revealed a two-phase learning pattern in models: an initial phase where specific features of the training data are learned, followed by a later phase where these representations are refined and abstracted for generalisation. In contrast, humans seem to bypass this staged process, directly forming generalizable representations from the onset of training, allowing for robustness against non-diagnostic features while retaining diagnostic attributes, and enabling recognition across identity-preserving transformations. Remarkably, the hurdle of generalisation lag persists not just within classic CNN frameworks but also in more recent architectures such as ViTs, suggesting a fundamental challenge in the alignment of generalisation behavior.

While shedding light on representational differences and learning dynamics, the presented work is accompanied by certain limitations. The scope of models analyzed represents only a small subset of available models and all employed models were pre-trained on ImageNet. Future studies might test and assess generalisation lag in models pre-trained on other, more human-like datasets such as SAYCam (Sullivan et al., 2021), or ecoset (Mehrer et al., 2021). Furthermore, also only a single dataset of novel objects was employed in the learning task. While we believe our stimuli choice should ensure external validity, it would be important to replicate our findings with other novel objects such Fribbles (Barry et al., 2014) or Greebles (Gauthier et al., 1998).To ensure the validity of the findings, future studies should also investigate whether the results persist under different hyperparameter settings and fine-tuning regimes.

## 5 CONCLUSION

Synchronizing learning conditions and comparing the full learning process revealed that DNNs surprisingly match human data efficiency, while also highlighting representational differences between humans and DNNs in terms of how quickly newly acquired knowledge can be generalised. We believe this insight to contribute to a deeper understanding of human cognition and learning, adding yet another desideratum to the list of properties artificial systems should possess to be behaviorally aligned with us: Increasing representational alignment requires more than enhancing data efficiency—instead our results indicate that we need to also focus on teaching DNNs to acquire immediately generalizable representations.

### DATA AND CODE AVAILABILITY

All code and data is available from https://github.com/wichmann-lab/supervised-learning-dynamics.

### ACKNOWLEDGMENTS

The authors appreciate Nik Meili's valuable assistance in collecting data from human participants. Special thanks go to the members of the Wichmann Lab and the Brüders Retreat for insightful discussions and their constructive input.

Lukas S. Huber was funded by the Swiss National Science Foundation (project number: P000PS_214659). Felix A. Wichmann is a member of the Machine Learning Cluster of Excellence, funded by the Deutsche Forschungsgemeinschaft (DFG, German Research Foundation) under Germany's Excellence Strategy – EXC number 2064/1 – Project number 390727645.

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

## A   APPENDIX: DETAILS ON THE GENERATION OF NOVEL OBJECTS

All novel 3D objects, used as a basis for the stimuli in this study, were created with the `Digital Embryo Workshop (DEW)` written by Mark Brady and Dan Gu. Some details on the employed algorithms can be found in Mark Brady's PhD Thesis (Brady, 1999, Appendix A, p.211). Table 2 shows the parameter settings for the generation of the parents and the second generation. Note that the Hydro PCD and Hydro Interaction PCD parameters were always kept at zero and for each batch of objects we used a random seed.

Table 2: Parameter settings for 3D object generation with DEW. "PCD" is short for "programmed cell death".

| Parameter | Parent Gen. | Second Gen. |
|---|---|---|
| Threshold Growth | 6 | 3 |
| Interaction Growth | 6 | 0 |
| Shrinkage PCD | 4 | 0 |
| Shrinkage PCD Interaction | 6 | 2 |

## B   APPENDIX: TEST DATA OF INDIVIDUAL OBSERVERS

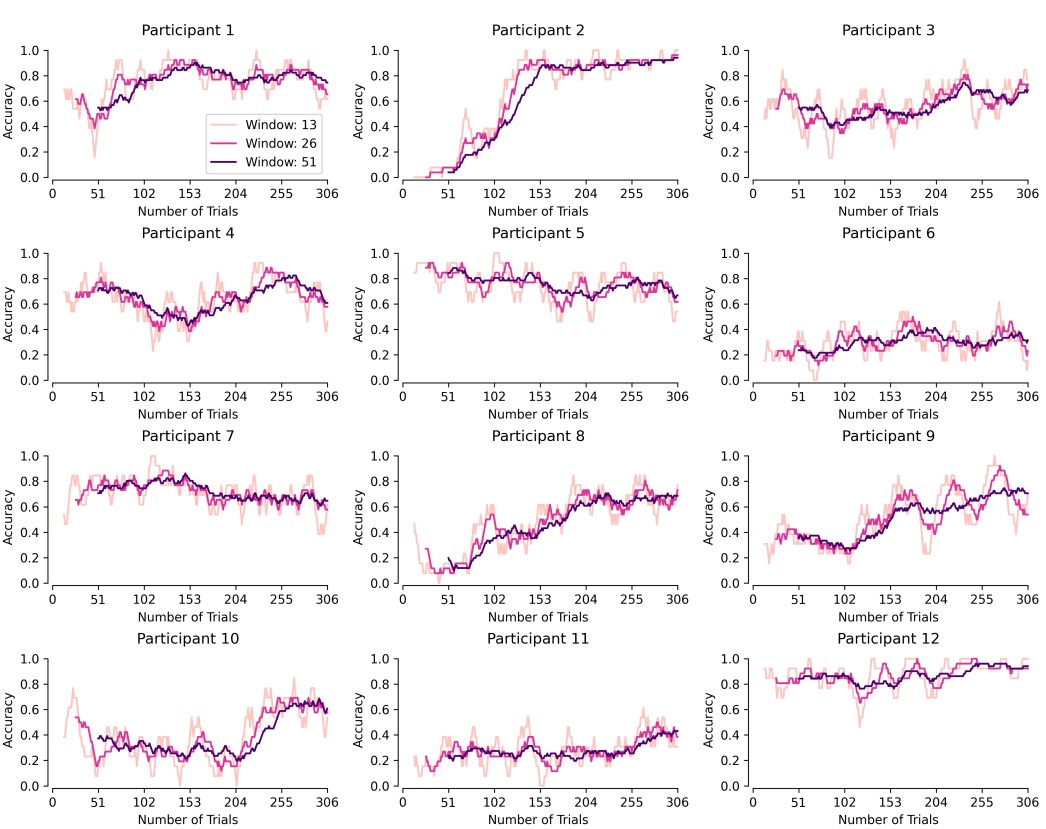

Figure 6: **Inter-individual differences in human test performance**. For each participant we show the evolution of test performance across the entire training period—depicted by moving averages of varying window sizes (different shades of pink). The largest window size of 51 equals the length of a test set.

## C   Appendix: A closer look at test performance

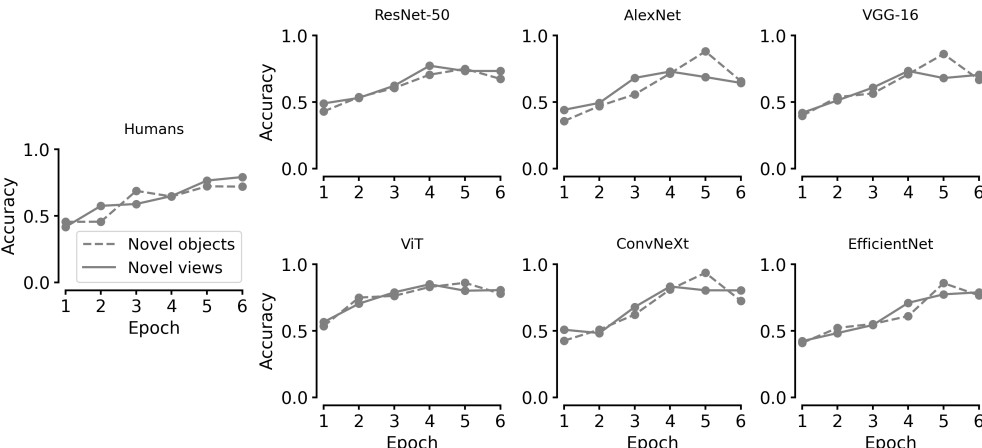

Figure 7: **Neither humans nor any model demonstrated a consistent advantage for either novel perspectives or novel objects.** All images were previously unseen; novel perspectives are viewpoints of training objects not encountered during training, while novel objects are entirely new and also not encountered during training. In each plot, test performance is divided into accuracy for novel perspectives (solid lines) and accuracy for novel objects (dashed lines).

## D  APPENDIX: DETAILS ON MODELS AND FINE-TUNING RUNS

All ImageNet pre-trained weights were obtained through the `PyTorch` model zoo (except the ViT which was obtained through the `Hugging Face` model library), where they are referred to as: "ResNet50_Weights.IMAGENET1K_V1" (ResNet-50), "AlexNet_Weights.IMAGENET1K_V1" (AlexNet), "VGG16_Weights.IMAGENET1K_V1" (VGG-16), "google/vit-base-patch16-224" (ViT), "ConvNeXt_Base_Weights.IMAGENET1K_V1" (ConvNeXt), and "Efficient-Net_V2_M_Weights.IMAGENET1K_V1" (EfficientNet). Figure 8 shows the results of the fine-tuning of those pre-trained model on our stimuli set.

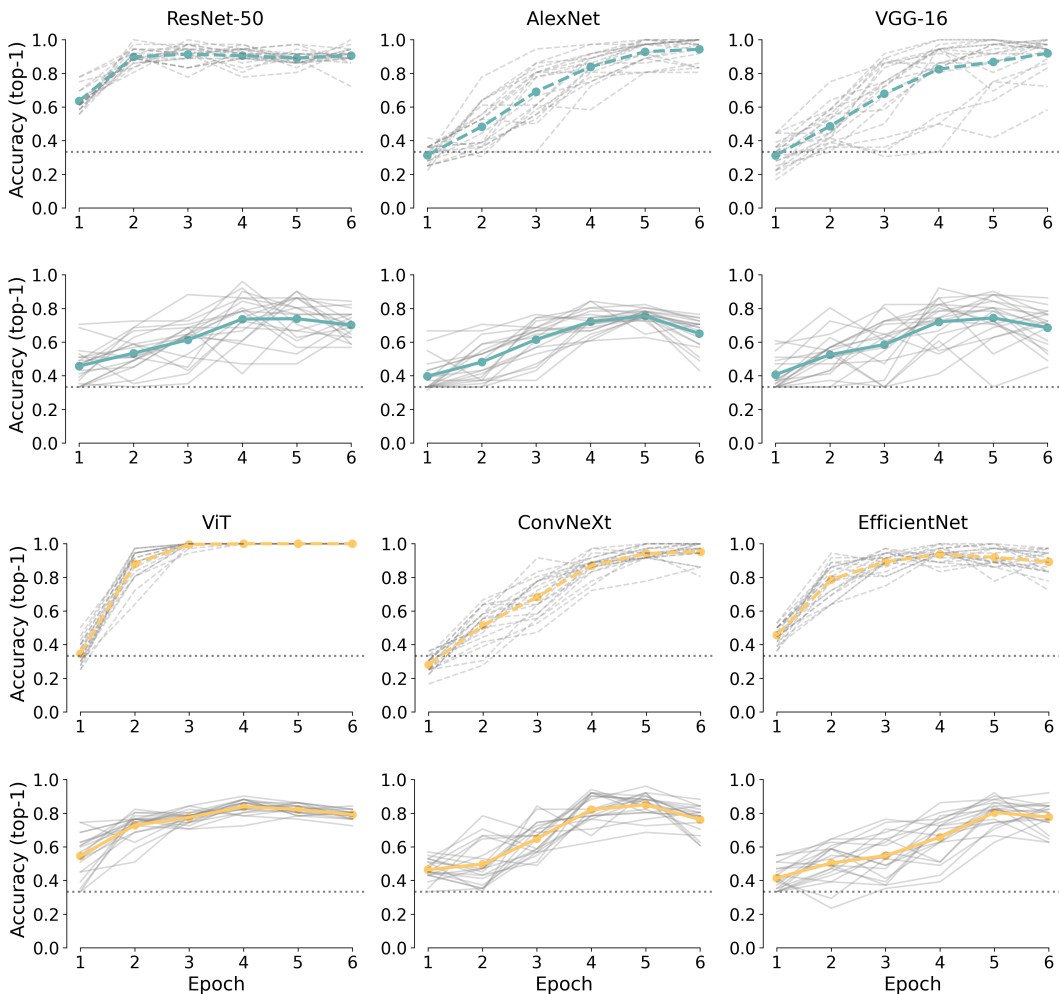

Figure 8: **Overview of all twenty fine-tuning runs for all classic CNNs (teal) and SOTA models (yellow).** Classification accuracy is plotted as a function of learning epochs for training (dashed lines) and test (solid lines) trajectories. Subtle grey lines indicate individual fine-tuning runs and colored trajectories inidcates average performance as reported in Figure 3. The large variation across fine-tuning runs seems not surprising given the small size of the training dataset (36 images) and might be attributed to the stochastic nature of the Adam optimizer, leading to different convergence paths and solutions.

