# OpenReview forum: "Comparing supervised learning dynamics: Deep neural networks match human data efficiency but show a generalisation lag"
_ICLR.cc/2024/Workshop/Re-Align — ICLR 2024 Workshop Re-Align Poster_

### Official Review · Reviewer_jdBS · 2024-02-24

**Rating:** 2
**Fit:** 3
**Confidence:** 2

**Workshop Review:**

This paper studies the representational alignment between humans and DNNs throughout training, rather than after training. It attempts to match the conditions between humans and DNNs as much as possible, and reveals that humans can generalize to held-out examples immediately, whereas DNNs first achieve high accuracy on training data before beginning to achieve nontrivial accuracy on held-out data. This work is presented well and studies the training dynamics of representational alignment, which is a fascinating question. Though some of the conclusions are unsurprising, I believe this work meets the bar for acceptance into the Re-Align workshop.


Strengths:
This paper studies an important and challenging question: how do the training dynamics of DNNs and humans compare to one another? Their stimulus creation and experimental design are rigorous and well-motivated, and they abide by many of the principles espoused by experts in human-machine comparisons (https://www.pnas.org/doi/pdf/10.1073/pnas.1905334117). They also study a variety of models, increasing the generality and relevance of their comparisons. Finally, they reveal an interesting generalization lag in DNN models, which is not present in humans.

Weaknesses:
Though the design is interesting, some of the conclusions are a bit muted. For example, it is clear that finetuning large models on such a small dataset will yield very high training accuracy, perhaps due to overfitting. In light of this, it is also somewhat unsurprising that there is a gap between achieving good accuracy on training examples and achieving good accuracy on held-out examples. The model is most likely starting out by memorizing the very small training set, before it actually adopts features that are useful for classification in general. A deeper analysis of the features learned after each epoch would clarify these dynamics.

**Reason For Not Giving Higher Score:**

The results are fairly unsurprising, and the analysis of the models after each epoch could be stronger.

**Reason For Not Giving Lower Score:**

The experimental design and stimuli generation is rigorous, and the work tackles an important (and very hard to address) question: how do the representations learned by humans and DNNs change over the course of training?

**Reviewer Domain:**

machine learning

---

### Official Review · Reviewer_v5St · 2024-02-24
**Evaluating generalization of humans and DNNs**

**Rating:** 2
**Fit:** 3
**Confidence:** 2

**Workshop Review:**

This paper presents an exploration of the immediate generalization capabilities of humans in contrast to DNNs, focusing on the domain of visual object recognition. The methodology is thorough, employing a variety of tasks and conditions to probe the nuances of human and ML and generalization.

**Reason For Not Giving Higher Score:**

While the comparison between human and DNN performance is insightful, the paper could benefit from a deeper analysis of why DNNs fail to generalize in the same ways humans do, possibly by including additional machine learning models (e.g. self-supervised architectures).

Other papers to include in the references would be:
1. Lake, B. M., Ullman, T. D., Tenenbaum, J. B., & Gershman, S. J. (2017). Building machines that learn and think like people. Behavioral and Brain Sciences, 40, E253.
2. Lee, M. J., & DiCarlo, J. J. (2023). How well do rudimentary plasticity rules predict adult visual object learning? PLOS Computational Biology, 19(12):e1011713.

**Reason For Not Giving Lower Score:**

1. The experiments are well-constructed, with logical progression and rigorous statistical analysis that supports the paper's conclusions.
2. The topic is highly relevant to cognitive science and ML communities, addressing a gap in our understanding of human-like machine learning.

**Reviewer Domain:**

machine learning

---

### Official Review · Reviewer_xfeJ · 2024-02-25
**It is challenging (maybe impossible) to adequately control for differences in learning of ANNs and Humans**

**Rating:** 1
**Fit:** 2
**Confidence:** 3

**Workshop Review:**

This paper investigates how the behavioral accuracy of humans changes while the humans learn novel categories, and compares this with the accuracy of neural networks when exposed to the same number of trials. This paper is motivated by the line of work comparing the behavior of deep neural networks to humans, and highlights that much of the previous work investigates representations of a network after training rather than during the training process.

Although the overall idea of the paper is interesting (to investigate how the learning process is performed in deep neural networks and human observers), the paper generally does not deliver on its promise to investigate how “transferable representations are acquired in human observers and various classic and state-of-the-art DNNs”, and I believe that some of the logic, experiments, analyses are flawed or insufficient to get at the question asked. Many more things than what are investigated in the paper would need to be controlled to say something interesting about the differences between human and machine learning of new objects. Due to this, I recommend rejection for the workshop at this time, but I hope that the comments below can help strengthen this paper or future papers.

Major (reasons for recommending rejections for the workshop):
* The paper is trying to tackle a very difficult question of comparing the learning dynamics of DNNs to those of humans. However, many of the key ingredients in this are unexplored, such as the learning rate and learning algorithm that is used during training. I suspect that some, if not all, of the resulting comparisons with humans could be easily modified with adjustments to to the learning rate, optimization algorithm, or loss function. Additionally, I am skeptical that this question is well-formed without major changes to make the learning algorithm itself biologically plausible.
* There is a large body of work in few-shot-learning and RL tackling the problem of comparing human learning to ANN learning, but references to this work are relatively untouched in the paper. Much of this other work tackles the problem by varying learning rate, learning algorithm, dynamics, curriculum learning, etc. Additionally, there is classic work on how humans learn new categories which isn’t sufficiently covered. Typically, missing references would not be a “Major” problem, but in this case one of my main criticisms is lack of novelty of experiments (even though it is possible that this *exact* comparison hasn’t been done, many similar ones have been, and unfortunately I am unconvinced that the main results of the paper teach us anything new).
* The statement about “absolute classification performance” being higher in DNNs compared to humans on the training data performance is unsurprising. In many training regimes (especially when the number of samples is small, such as here), DNNs are known to memorize the training data, especially when the number of training samples is small (which differs from humans, but that is well reported and is the main motivation for one and few-shot learning experiments). Also, human observers are known to exhibit attentional lapses during behavioral tasks which can lead to lower performance than ANNs, which should at minimum be commented on as one of the reasons for lower accuracy, and ideally would be regularly tested during the experiment with catch trials.
* Although the introduction of the paper is motivated by “representational alignment” between the models and humans, the measure of representation that the authors use, overall accuracy on the task, does not seem to fit into a general definition of “representation”, even with the idea of making this as lenient as possible for the purposes of the workshop. I was expecting something that could give insight into internal representations, such as patterns of errors, but this was not explored.
* The participant exclusion criteria seem to have been determined post hoc in a way that potentially biases the results. Exclusions should ideally be done using separate data from that used in the analysis to avoid cherry picking participants that tell the correct story. I was particularly confused about the exclusions of Participant 3, 4 and Participant 5, as they seemed to perform significantly above chance on the task (which would be indicative of learning) but learned it too quickly to be included in the analysis. The inclusion of such participants would potentially make the curves in Figure 3 higher for human observers, as these participants are demonstrating very “data-efficient” learning.
* I have concerns about the train-test split of the data. Specifically, it sounds like *images* were used for splitting the data and not specific *objects*? It seems a bit non-standard to include *exactly* the same images for training each time. What happens when different images of the same object are used instead, and generalization is only over novel objects in the same class? Additionally, were precautions taken to make sure the same objects were not used in the train and test sets?

Minor (would improve the paper):
•	I generally disagree with the statement that ImageNet training results in models that are approximately equivalent to the existing capacity of humans for image classification accuracy (page 2 “starting point” and referenced in line 4 of the discussion). I suggest revising this motivation, especially because the images tested are out of distribution for ImageNet.
•	The idea of development and adult learning is conflated throughout the paper. These are two distinct learning processes and should be covered with appropriate background on each (changing citations where appropriate).
•	How was the model modified to enable predictions across only 3 tasks? I could not find this detail.

**Reason For Not Giving Higher Score:**

The exclusion of participants in behavioral analysis seems suspicious. Appropriate controls were not performed to modify the learning dynamics of the DNNs. Major lack of reference to previous literature on the topic of biological vs. artificial learning.

**Reason For Not Giving Lower Score:**

The "representation" compared in the paper was the overall accuracy at various points in training. In my mind, this does not count as a topical fit for the workshop, but perhaps others would disagree.

**Reviewer Domain:**

cognitive science

---

### Decision · Program_Chairs · 2024-03-02

Accept (Poster)